# Comprehensive Monitoring Method for Diaphragm Wall Deformation Combining Distributed and Point Monitoring in Key Areas

**DOI:** 10.3390/s25072232

**Published:** 2025-04-02

**Authors:** Chun Lan, Hui Zhang, Guangqing Hu, Feng Han, Heming Han

**Affiliations:** 1Anhui Province Green Mine Engineering Research Center, Hefei 230088, China; lansedesha-100@163.com (C.L.); huguangqing117@ustc.edu (G.H.); 2Anhui Province Land Space Planning Research Institute, Hefei 230601, China; 3Anhui Province Land Development Reclamation Consolidation Center, Hefei 230601, China; 4College of Resources and Environment, Anhui Agricultural University, Hefei 230036, China; 13678628735@163.com; 5Exploration Research Institute, Anhui Provincial Bureau of Coal Geology, Hefei 230088, China; 6School of Resources and Environmental Engineering, Hefei University of Technology, Hefei 230009, China; hanheming@hfut.edu.cn

**Keywords:** fiber optic monitoring, deep displacement monitoring, ultra-weak fiber Bragg grating, diaphragm wall, deflection calculation

## Abstract

The diaphragm wall plays an important role in the safe construction of foundation pits, and it is crucial to accurately monitor its deformation in real time. Traditional monitoring methods often face challenges in achieving distributed monitoring, and the cost of using fiber optic sensors for real-time and distributed monitoring can be prohibitively high. To improve the monitoring efficiency and accuracy of the deep deformation of the diaphragm wall, this paper proposes a hybrid monitoring method that combines ultra-weak fiber Bragg grating (UWFBG) technology and traditional FBG sensors. This distributed–discrete optical fiber monitoring approach allows for continuous, high-resolution data collection along the diaphragm wall while providing targeted, real-time measurements at critical locations. Fiber optic crack testing of concrete beam structures was carried out to verify the method of evaluating the health status of structures using distributed fiber optic data. An engineering case study was developed to validate the feasibility of this method. The results demonstrated that the hybrid approach effectively captures the overall deformation distribution of the diaphragm wall while enabling real-time monitoring of key areas, including the detection of crack initiation and propagation. The proposed method offers a significant advancement in deformation monitoring, providing enhanced accuracy, spatial coverage, and the ability to detect both macro-scale trends and micro-scale anomalies, which is particularly beneficial for complex underground structures.

## 1. Introduction

With the increasing demand for underground space development, foundation pit projects are increasingly characterized by greater excavation depths and more complex geological conditions [1,2,3,4]. Consequently, ensuring the safety of foundation pit construction has become a critical concern. The instability or failure of retaining structures within foundation pits can result in significant casualties and substantial economic losses [5,6]. Therefore, it is essential to focus on the precise and efficient monitoring of the deformation of supporting structures. This is particularly important with the rising use of diaphragm walls, where the monitoring and collection of deformation data from deep within the wall has garnered increasing attention [7,8]. Consequently, the development of methods for monitoring deep displacements in diaphragm walls is of paramount importance.

Traditional displacement monitoring methods, such as theodolites, inclinometers, and laser rangefinders, have inherent limitations in terms of accuracy, efficiency, and resistance to interference, which makes them insufficient to meet the demands of modern monitoring [9,10]. In contrast, optical fiber sensing technology offers several advantages, including high precision, long-distance capabilities, electromagnetic immunity properties, and the ability to provide distributed monitoring [11,12,13]. Broadly, optical fiber monitoring can be classified into quasi-distributed techniques, such as fiber Bragg grating (FBG), and fully distributed technologies like BOTDR/BOTDA [14,15]. However, fully distributed optical fiber systems typically require expensive demodulation equipment, and achieving both distributed and real-time monitoring of underground continuous walls presents significant challenges. Among these technologies, ultra-weak fiber Bragg grating (UWFBG) stands out due to its exceptional sensitivity, high precision, real-time monitoring capabilities, and distributed nature [16], making it an ideal solution for deep displacement monitoring in diaphragm walls. The rapid development of fiber optic sensing technology has revolutionized structural health monitoring (SHM) in geotechnical engineering, providing unprecedented capabilities in real time and distributed and high-precision measurements. In the past decade, advancements in fiber Bragg grating (FBG) and distributed fiber optic sensing (DFOS) technologies have addressed key challenges in monitoring underground structures, slopes, tunnels, and retaining walls [17,18].

To address the issues of the insufficient measurement accuracy, low efficiency, and poor electromagnetic immunity ability of traditional monitoring methods, this paper proposes a hybrid monitoring approach that combines UWFBG technology with traditional FBG technology. This method achieves a balance between distributed coverage and point-based accuracy, solving the shortcomings of traditional point-based monitoring, which has blind spots, and the limitation that distributed monitoring is difficult to economically monitor key areas in real time. The feasibility of this method is investigated through both the theoretical analysis and fiber optic crack testing of concrete beam structures. An engineering case study is also conducted to validate the performance of the hybrid monitoring system. This method provided a promising solution for the safe and reliable construction of complex underground structures.

## 2. Monitoring Principle of Diaphragm Wall with Fiber Optic Sensing

### 2.1. FBG Sensing Principle

Fiber Bragg grating (FBG) is created by exposing the core of a single-mode optical fiber to strong ultraviolet (UV) light with a periodic pattern. The UV exposure permanently increases the refractive index of the fiber core, generating a fixed refractive index modulation according to the exposure pattern. This fixed refractive index modulation is referred to as the grating.

At each spatial periodicity of the refractive index change, a small portion of the light undergoes reflection. When the grating period is approximately half of the incident light’s wavelength, all reflected light coherently combines into a beam with a specific wavelength, resulting in a large reflection. This is known as the Bragg condition. The wavelength at which the incident light undergoes reflection is called the Bragg wavelength. The light of the other wavelengths is almost unaffected by the Bragg grating and will continue to transmit through the optical fiber grating. For an illustration of this principle, please refer to Figure 1.

Therefore, when light propagates through the grating, there is almost no signal attenuation or change. Only wavelengths that satisfy the Bragg condition will be affected and cause strong reflection. The precise setting and maintenance of the grating wavelength’s performance is a fundamental feature and advantage of fiber Bragg grating (FBG).

The center wavelength of the reflected light satisfies the following Bragg equation:(1)$$\lambda_{B\mathrm{ragg}}=2n\Lambda$$
where *n* is the refractive index, and Λ is the grating period.

Since the refractive index and grating period are affected by temperature and strain, the center wavelength of the Bragg reflected light will also change with variations in temperature, strain, or both, as shown in Figure 2. Therefore, by measuring the change in the center wavelength of the reflected light, the corresponding physical quantity variation at the monitored location can be determined.

Based on the FBG technology principle, high-precision point-based monitoring can be achieved for key areas [19]. Strain data can be used to identify cracks and assess the effectiveness of supporting structures.

### 2.2. Ultra-Weak FBG Technology

Ultra-weak fiber Bragg grating UWFBG) technology uses low-reflectivity gratings (typically < −20 dB) to enable multiple gratings to overlap without interference, as shown in Figure 3. Time-division multiplexing (TDM) allows monitoring of several locations along a single fiber, making UWFBG ideal for long-range, real-time monitoring in applications like foundation pit monitoring [20].

The two electro-optic modulators (EOMs) serve distinct roles: the first generates narrow optical pulses for spatial resolution enhancement, while the second modulates wavelength scanning using a tunable laser (C-band, 1525–1565 nm). The system operates at a repetition rate of 10 kHz and an acquisition rate of 1 Hz, ensuring real-time strain monitoring. All gratings are photowritten at the same nominal wavelength (1550 nm), minimizing spectral overlap and enabling precise detection of strain peaks through wavelength shift analysis.

Signal demodulation is achieved using a tunable laser, electro-optic modulator, and erbium-doped fiber amplifier, with reflected light collected by a photodetector. Shifts in the reflected wavelength indicate changes in strain or temperature, with temperature compensation ensuring accurate strain measurements [21,22].

UWFBG technology offers high sensitivity, distributed sensing, and immunity to electromagnetic interference, along with long-term stability and low power consumption, making it ideal for large-scale, long-term monitoring.

## 3. Distributed-Point Comprehensive Deformation Monitoring of Diaphragm Wall

The diaphragm wall is loaded by the soil outside the foundation pit. The wall will undergo bending deformation. The bending deformation of the diaphragm wall is regarded as pure bending. The longitudinal linear strain at the position of *z* from the neutral axis on the cross-section is investigated.(2)$$\varepsilon =z\frac{d\alpha }{dx}=\frac{z}{r}$$
where *r* is the radius of curvature; *α* is the angular displacement relative to the original position of the cross-section.

In mathematics, the relationship between the curvature of a plane curve and the derivative of a curve equation is as follows:(3)$$\frac{1}{r(x)}=\pm \frac{w^{\prime \prime }(x)}{{\left(1+w^{\prime }(x)\right)}^{3/2}}$$
where *w*(*x*) is the deflection curve equation.

The following can be seen from Equation (1):(4)$$\varepsilon (x)=\frac{z}{r(x)}$$

As shown in Figure 4, high-precision point-type FBG inclinometers are deployed in the middle and lower part of the single pile to achieve accurate monitoring of key areas of the ground-connected wall. The displacement distribution of the ground-connected wall can be calculated by laying sensor optical cables on the ground-connected wall. In this way, the overall deformation law of the ground-connected wall and the fine deformation state of key areas can be fully understood.

The strain peak can be regarded as a sign of crack generation (Figure 5). The peak point of this strain peak represents the location where the crack is generated. The width of this crack can be obtained by integrating the strain peak.

## 4. Tests

### 4.1. Test Equipment

Since high-strength optical cables are often used in actual projects to monitor structural deformation, and such optical cables are insensitive to strain peaks, this paper uses 0.9 mm optical cables to monitor the crack development law of concrete beams and compares and analyzes the strain level of high-strength optical cables at different crack development levels, providing a basis for the evaluation of the health status of real engineering structures.

As shown in Figure 6, the concrete beam loading device consists of three parts: bracket, distribution beam, and jack. A 0.9 mm tight-buffered optical cable and metal-based cable-shaped strain sensing optical cable are laid at the same position on the concrete beam surface. A concrete beam is placed on the bracket, a distribution beam is placed in the middle of the beam, and a jack is placed on the distribution beam. When loading, the jack is pushed upward to generate a downward force, which is divided into two downward components by the distribution beam and acts on the middle of the beam. During the test, the initial load is 0 kN, and the load on the beam is increased by 2 kN at each level. After each level of load is applied, the test begins after it stabilizes. The test ends when the load is applied until the sensing optical cable or sensor is damaged. In addition, the cracks on the surface of the concrete beam are read by the crack width gauge to compare with the fiber optic monitoring data.

### 4.2. Test Results Analysis

As shown in Figure 7, each corresponding load to crack initiation is recorded. Crack progression was quantified by integrating strain peaks into widths, validated against crack width gauge measurements, with faster propagation rates observed in high-stress zones under distribution beams.

The strain data show that 12 obvious strain peaks were monitored during the loading test, indicating that 12 cracks were generated inside the concrete beam. The crack widths are shown in Table 1. The cracks are evenly distributed, and the spacing between adjacent cracks is about 10 cm. The cracks at different positions in the horizontal direction have different crack-widening rates as the load increases. The relationship between different crack expansion rates and loads is shown in Figure 7. It can be seen from the figure that as the load increases, the crack expansion rate is a constant increase, and the crack expansion rates at different positions are different. Cracks 4, 5, and 9 have the fastest expansion rates, close to 0.007 mm/kN. Cracks 1 and 12 are located on both sides of the beam, and the crack expansion rate is the smallest, less than 0.002 mm/kN. The crack expansion rates at cracks 4 and 9 are the fastest because these two places are the locations where the distribution beams are erected, the forces they receive are large, and the bending moments they generate are also the largest, so the strain expansion rate is also the largest, and the crack development is also the largest. Cracks 1 and 12 receive relatively small forces, and the bending moments they generate are also relatively small, so the crack expansion rates are also relatively small.

As shown in Figure 8, the strain peak generated by the steel strand cable is not obvious, but the location of the crack can still be detected, and the strain peak range of the crack is larger. Before 36 kN, about 1600 micro-strains were generated, and the maximum crack was 0.23 mm. This shows that due to the large elastic modulus of the steel strand cable, its strain isolation is also relatively large. When the crack width expands to 0.23 mm, the adjacent cracks cannot be identified.

As the load increases, the crack width shows a significant nonlinear trend. Initially (0–10 kN), the crack width increases slowly, but beyond approximately 20 kN, the rate of increase accelerates, especially above 50 kN, where the crack width grows sharply (Figure 9). This indicates that the structure primarily exhibits elastic deformation at lower loads, while plastic deformation or crack propagation may occur at higher loads. Therefore, monitoring should be intensified above 20 kN, and loads exceeding 50 kN should be considered as a warning threshold, requiring prompt action to prevent structural instability or failure. The strain level of the high-strength strain optical cable corresponding to a load of 20 kN is 800 microstrain, which provides a basis for the subsequent health evaluation of the ground-connected wall.

## 5. Case Study

### 5.1. Overview

The study area is located in Baoshan District, Shanghai, and the Yangtze River south road section of Shanghai Metro line 18 is to be built. The survey revealed that the foundation soil of the station is within the depth of 75.0 m, all of which are loose Quaternary sediments, belonging to the Quaternary coastal, estuary, shallow sea, swamp, drowned valley, and lacustrine facies sedimentary layers, mainly composed of saturated sediments. It is composed of cohesive soil, silty soil, and sandy soil and generally has the characteristics of layered distribution. Figure 10 shows the mechanical parameters of the soil. The CX7 optical fiber monitoring point in the foundation pit was introduced. The depth of the diaphragm wall is 30 m, and the estimated excavation depth is 18 m. Five concrete struts are presented. Based on the empirical analysis, the tilt sensor installation points were selected as 7 m and 25 m.

### 5.2. Installation of Optical Fiber Sensor

As shown in Figure 11a, the high-strength strain optical cable used in this project has good mechanical properties and tensile and compressive properties and can be well coupled with rock, soil, concrete, and other structures (the same model as used in indoor tests). The construction is convenient, and, at the same time, it can resist all kinds of bad working conditions. The demodulator can collect thousands of points simultaneously, and the system integration is high. Figure 11b shows the figure of demodulation equipment.

In the process of processing the reinforcement cage of the diaphragm wall, the optical cable laying line is determined (Figure 12a), and the ultra-weak FBG and tilt sensor are bound at the design position of the reinforcement cage (Figure 12b). When selecting the layout position, it is necessary to avoid welding, oxygen cutting, and conduit cabin, which are easy to damage the optical cable. In the process of lowering the reinforcement cage, the construction personnel guard the whole process to prevent the reinforcement cage from being damaged in the process of lowering (Figure 12c). Finally, the top head of the diaphragm wall is protected, and the initial value is obtained by using a portable optical fiber demodulator (Figure 12d).

### 5.3. Discrete Point Monitoring Results

The angular displacement data of the diaphragm wall, as shown in the Figure 13, illustrates distinct deformation patterns. Notably, deformation is more pronounced in the lower sections of the wall, with a decreasing trend toward the upper regions, suggesting a depth-dependent response to the increased lateral pressure at greater depths. Localized clusters of higher displacement indicate potential zones of concentrated deformation, which may reflect structural irregularities, variations in soil properties, or excavation-related disturbances. These trends underscore the importance of spatially varying soil–structure interactions and highlight the need for further analysis of the underlying factors contributing to the observed deformation. Overall, the data provide valuable insights into the wall’s stability, with implications for evaluating potential risks and optimizing excavation practices.

As shown in Figure 14, the time-series data of angular displacement for the diaphragm wall reveal a gradual increase in deformation over time, corresponding to the progressive excavation and increasing soil pressure. This is followed by a phase of stabilization, indicating that the wall has reached an equilibrium state in response to the excavation. However, short-term fluctuations in displacement are observed, likely due to transient factors such as groundwater variations or temporary construction disturbances. Notably, abrupt peaks in displacement may indicate sudden load changes or localized structural responses. These deformation patterns, when correlated with key excavation milestones, provide critical insights into the dynamic soil–structure interaction and highlight the importance of continuous monitoring to assess stability and optimize construction practices.

### 5.4. Analysis of the Strain–Depth Curve

The strain–depth curve revealed distinct deformation patterns by identifying strain concentration zones, enabling division of the wall into critical regions like the upper resistance area (URA) and main load area (MLA). This analysis clarified how excavation stages and strut placements influenced strain distribution, providing insights into crack risks and structural stability.

As shown in Figure 15, the strain distribution outside the diaphragm wall was positive–negative. At the initial stage of excavation, because the top strut of the diaphragm wall has not been formed, the strain of the diaphragm wall is negative. When the top strut can provide a certain support reaction, the upper strain of the diaphragm wall is positive, and the lower strain is negative. Therefore, according to the strain characteristics, the diaphragm wall was divided into different areas (Figure 15): upper resistance area (URA) and main load area (MLA). For the upper resistance zone, the strain changes from negative to positive with the increase in excavation depth. Furthermore, stress concentration shows in the upper part of the second strut. For the main load area, the deformation increased with the progress of excavation, and the strain peak appeared at a depth of 11 m. When the excavation depth increases, the struts have limited influence, and the local strain peak appears. The horizontal displacement of the middle part of the wall was triggered by the active earth pressure, while the upper support and the soil below the excavation face hindered the wall movement. Based on the experimental analysis of the beam structure, it is known that when the strain level measured by the optical cable reaches an 800 microstrain, it is necessary to focus on the analysis of the obvious cracks that may occur in the structure. Based on the on-site monitoring data, it can be seen that the maximum strain of the underground continuous wall is less than a 300 microstrain, so there may be certain tiny cracks in the structure, but the overall structure still maintains a linear deformation stage and can provide good support.

## 6. Discussion

The spatial resolution of the optical cable actually used on site in this paper is 1 m, which can achieve the comparison of relative strain values within the 2 m length range of the structure. This can be unified with the strain distribution range of the beam in the indoor test. In addition, this paper uses optical cables sensitive to local cracks and high-strength optical cables (the same indoor and on-site) for the test results of the same quantity and establishes the relationship between the strain level obtained by the high-strength optical cable and the crack appearance and development of the crack-sensitive optical cable. Finally, the strain data of the high-strength optical cable of the on-site ground-connected wall can be used to evaluate the local health status of the structure. In addition, the horizontal movement of the ground-connected wall causes possible accidents, and this paper uses FBG inclinometer data to achieve monitoring and evaluation.

This paper combines distributed strain measurement with local inclination measurement, uses UWFBG to obtain distributed strain data of underground continuous walls, and identifies local damage to the structure through strain singular points. The horizontal motion state of the wall is calculated through the data obtained from the FBG inclinometer array, which can comprehensively monitor the local damage and overall motion state of the underground continuous wall. Compared with traditional point sensors, this paper can obtain the deformation state of the structure in a distributed manner and realize the identification of local damage.

By merging continuous, high-resolution data from the distributed system with targeted, real-time measurements at critical locations, this method enhances both spatial coverage and precision. It allows for the detection of distributed deformation anomalies and provides real-time monitoring of localized structural concerns, improving the overall reliability and accuracy of structural health assessments. This integrated approach offers a comprehensive view of the wall’s deformation behavior, addressing both macro-scale trends and micro-scale anomalies. A key innovation of this hybrid method is its ability to effectively monitor cracks in supporting structures using the distributed strain measurement technology, which can capture subtle deformation changes along the entire length of the diaphragm wall. This capability provides a more detailed understanding of crack initiation and propagation, which is often difficult to detect with discrete sensors alone. It is particularly valuable for complex underground structures, where varying deformation patterns may arise due to factors such as soil heterogeneity, excavation stresses, or construction disturbances. The real-time data from discrete sensors enable the early detection of potential failure mechanisms, facilitating timely interventions. In addition, the crack development index has not yet established a quantitative relationship with the strain data obtained by the optical cable, which will be further studied in subsequent studies.

In terms of innovation, this method overcomes the limitations of traditional monitoring approaches that rely on either distributed or discrete sensors alone, offering a more robust and flexible solution with greater sensitivity to deformation. Future enhancements, such as incorporating advanced data processing and machine learning, could further refine anomaly detection and improve decision-making. While challenges remain in terms of cost, sensor calibration, and system integration, addressing these issues could broaden the applicability of this approach to other geotechnical structures, such as tunnels and dams.

In conclusion, the hybrid monitoring method presents a promising tool for effective, comprehensive deformation monitoring, improving structural integrity assessments, and risk management in geotechnical engineering.

## 7. Conclusions

An advanced method of monitoring the deep displacement of the diaphragm wall based on ultra-weak FBG and FBG was introduced in this paper. This paper proposes a distributed–discrete optical fiber monitoring method. The effectiveness of the method is verified by practical engineering applications. The following conclusions were obtained:(1)The ultra-weak FBG technology can accurately obtain the deep deformation information of the diaphragm wall. The technology can provide more efficient and accurate reference for the design and construction of diaphragm walls.(2)The steel strand-reinforced UWFBG cable used in the case has the advantages of simple and efficient installation. The modified fiber optic cable can be applied to difficult and complex conditions and has great application potential.(3)The proposed hybrid approach, combining fiber optic distributed monitoring with discrete point sensors, offers a significant advancement in the deformation monitoring of underground diaphragm walls. A key advantage of this approach is its ability to effectively monitor cracks in supporting structures using distributed strain measurement technology. This enables the detection of subtle deformation changes along the diaphragm wall, allowing for the early identification of crack initiation and propagation, which would be challenging to detect with discrete sensors alone. This innovation significantly enhances the monitoring capability by providing a more comprehensive and accurate understanding of structural health, especially in detecting micro-scale anomalies and localized deformations.

## Figures and Tables

**Figure 1 sensors-25-02232-f001:**
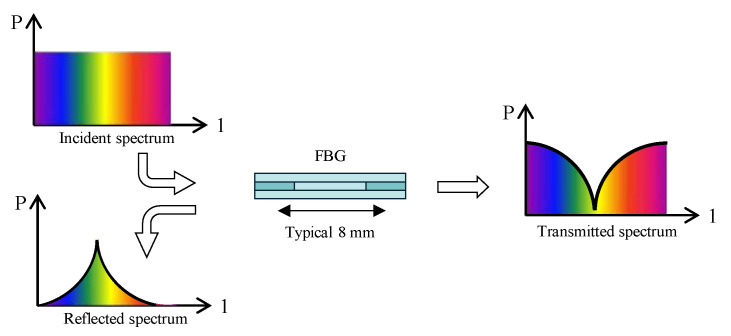
Working principle of FBG.

**Figure 2 sensors-25-02232-f002:**
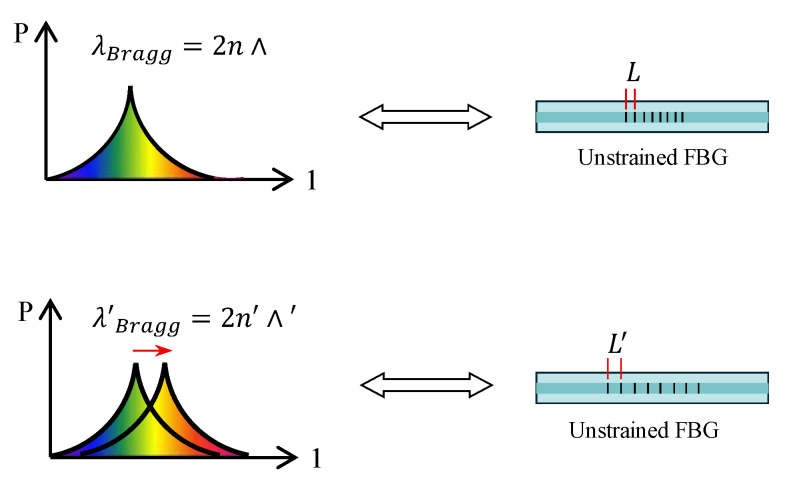
FBG response as function of strain.

**Figure 3 sensors-25-02232-f003:**
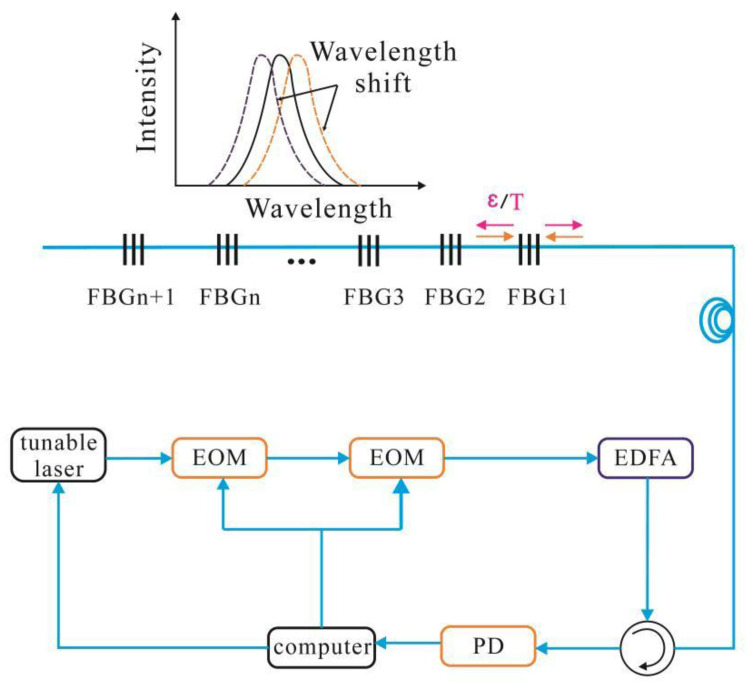
Sensing principle of UWFBG.

**Figure 4 sensors-25-02232-f004:**
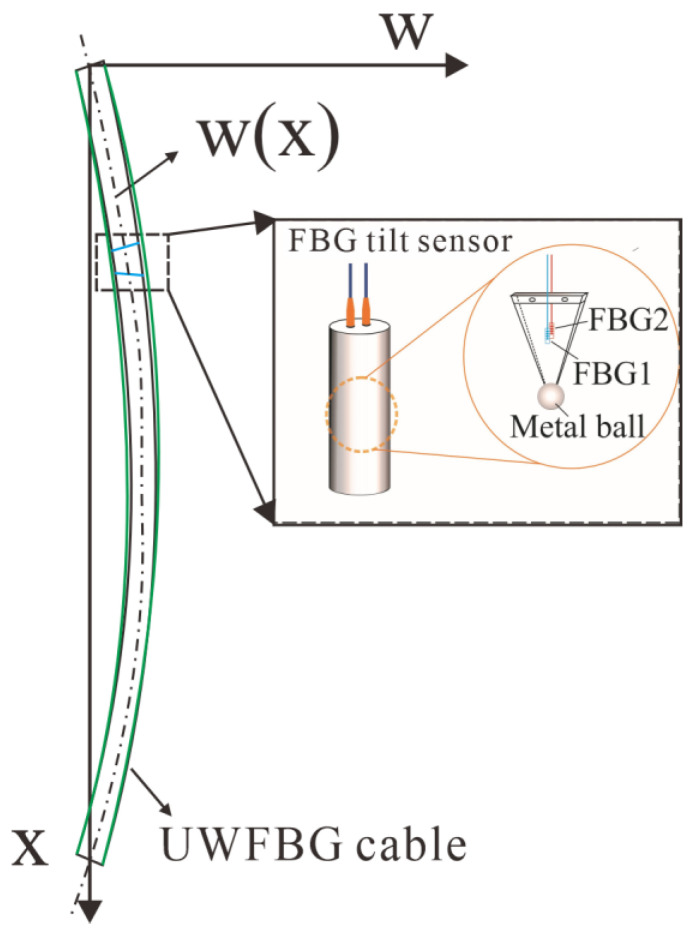
Schematic diagram of distributed-point comprehensive deformation monitoring.

**Figure 5 sensors-25-02232-f005:**
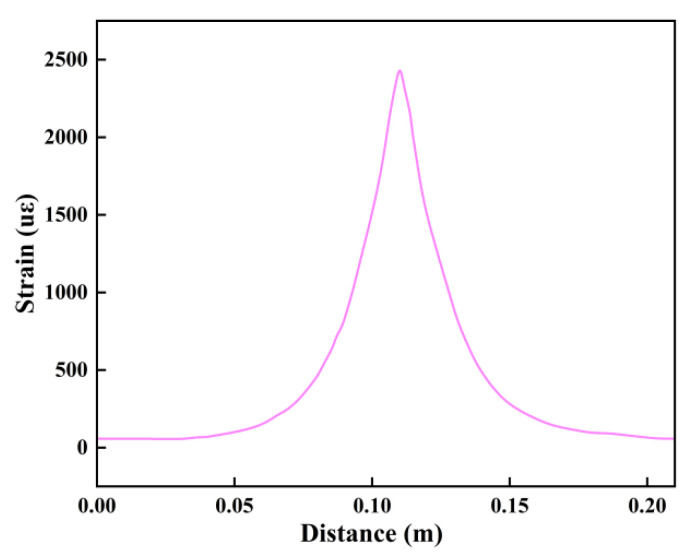
Schematic diagram of strain peaks caused by cracks on optical cables.

**Figure 6 sensors-25-02232-f006:**
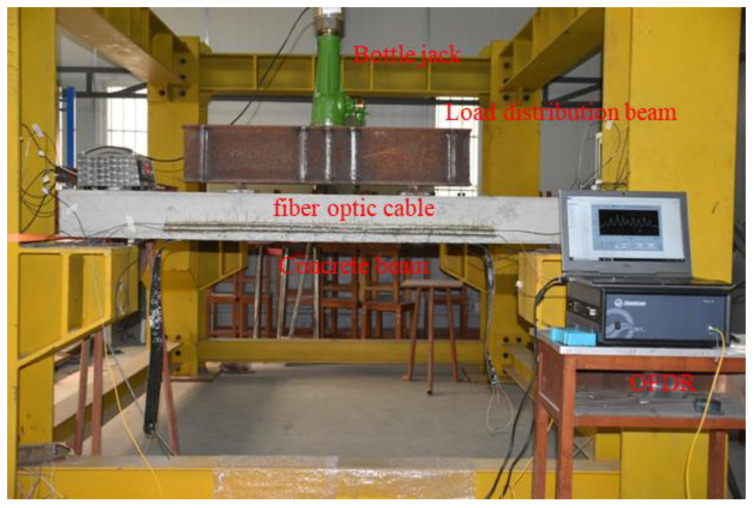
Concrete beam loading test.

**Figure 7 sensors-25-02232-f007:**
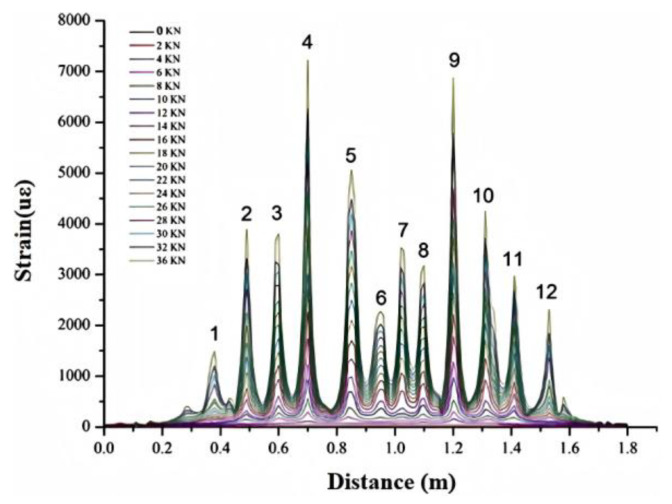
Monitoring results of 0.9 mm diameter optical cable on concrete beam cracks. (The number represents the crack number).

**Figure 8 sensors-25-02232-f008:**
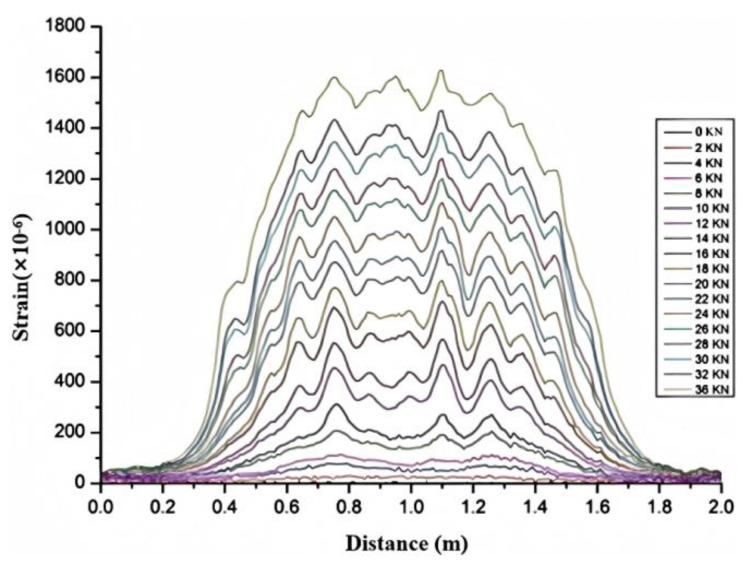
Strain monitoring results of high–strength optical cable in cracks on concrete beams.

**Figure 9 sensors-25-02232-f009:**
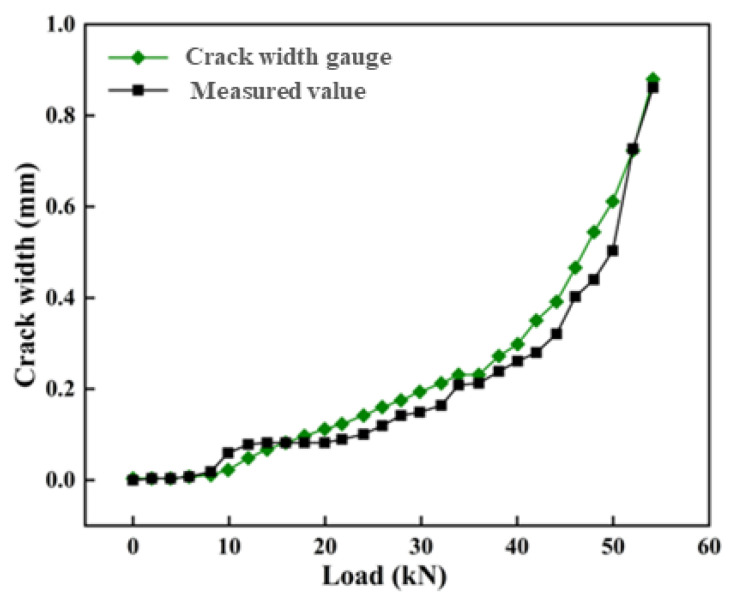
Comparison between the optical fiber monitoring crack value and the crack width gauge value.

**Figure 10 sensors-25-02232-f010:**
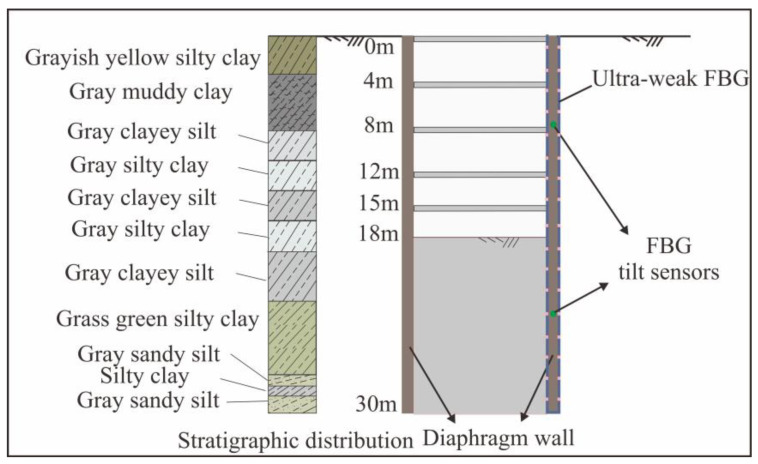
Soil distribution of foundation pit.

**Figure 11 sensors-25-02232-f011:**
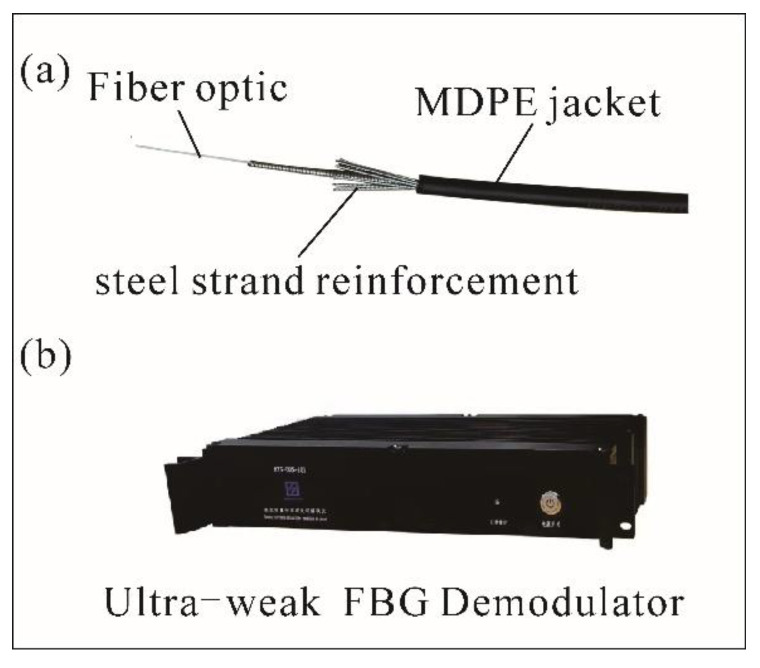
Physical picture of ultra-weak FBG and demodulation: (**a**) Fiber optic structure; (**b**) demodulation.

**Figure 12 sensors-25-02232-f012:**
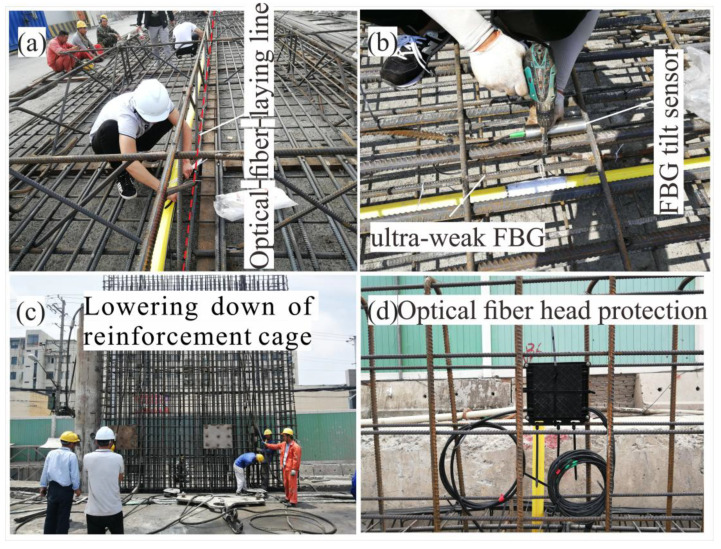
Field installation of optical fiber sensor: (**a**) optical-fiber-laying line; (**b**) installation of ultra-weak FBG and FBG tilt sensor; (**c**) lowering down of reinforcement cage; (**d**) optical fiber head protection.

**Figure 13 sensors-25-02232-f013:**
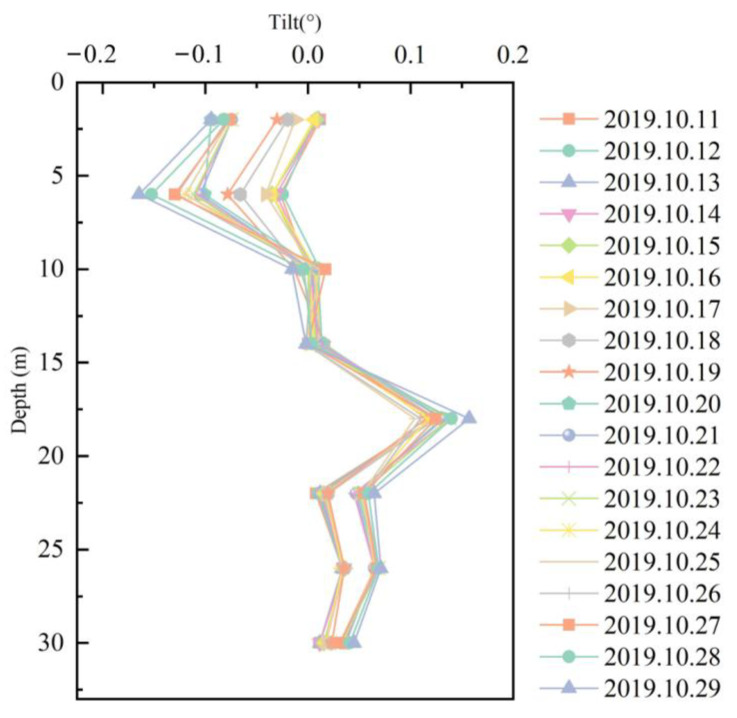
Discrete point monitoring results of diaphragm wall based on FBG sensors.

**Figure 14 sensors-25-02232-f014:**
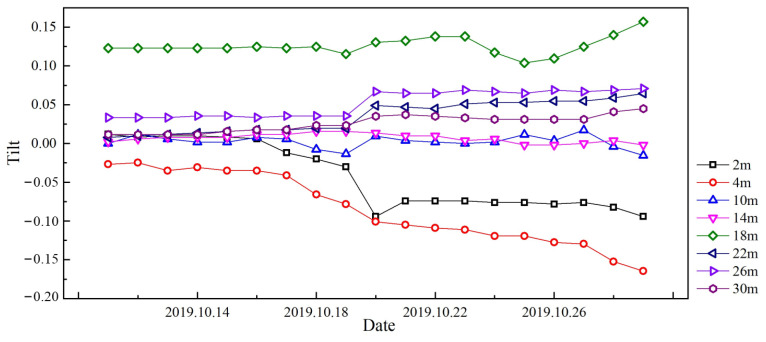
Tilt–time monitoring results of diaphragm wall based on FBG sensors.

**Figure 15 sensors-25-02232-f015:**
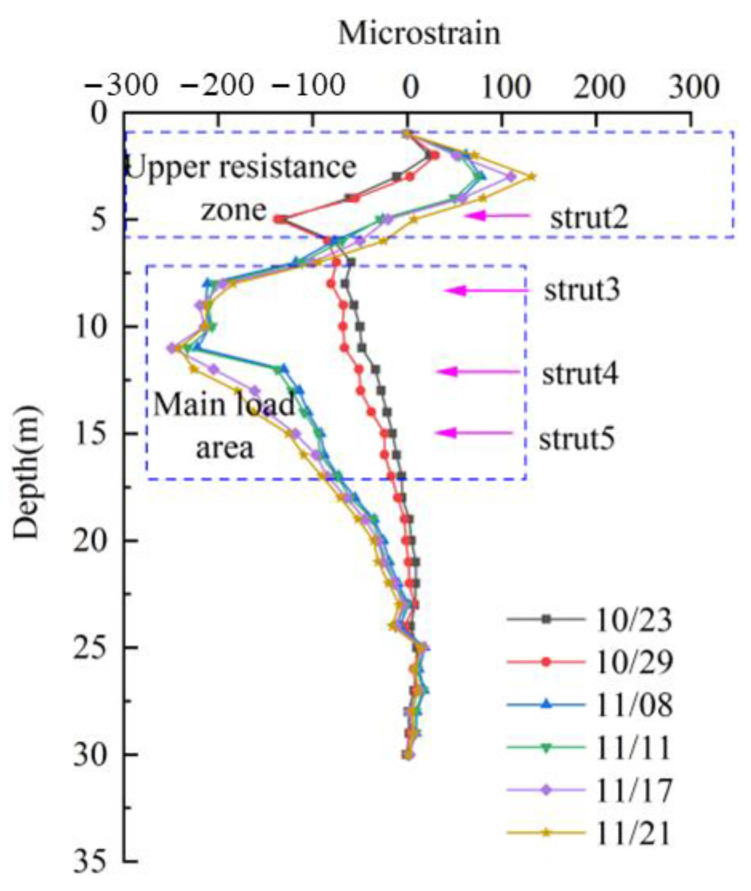
CX7 strain–depth curve based on UWFBG.

**Table 1 sensors-25-02232-t001:** The 0.9 mm tight-packed optical cable crack monitoring width.

Load/kN	Crack 1	Crack 2	Crack 3	Crack 4	Crack 5	Crack 6	Crack 7	Crack 8	Crack 9	Crack 10	Crack 11	Crack 12
0	0.00	0.00	0.00	0.00	0.00	0.00	0.00	0.00	0.00	0.00	0.00	0.00
2	0.00	0.00	0.00	0.00	0.00	0.00	0.00	0.00	0.00	0.00	0.00	0.00
4	0.00	0.00	0.01	0.01	0.01	0.01	0.00	0.01	0.01	0.01	0.00	0.00
6	0.00	0.01	0.01	0.02	0.01	0.01	0.01	0.01	0.02	0.01	0.00	0.00
8	0.00	0.01	0.02	0.02	0.02	0.01	0.01	0.01	0.03	0.02	0.01	0.01
10	0.01	0.02	0.02	0.04	0.04	0.02	0.01	0.02	0.04	0.02	0.02	0.01
12	0.01	0.02	0.03	0.05	0.05	0.03	0.02	0.02	0.05	0.03	0.02	0.01
14	0.01	0.03	0.04	0.06	0.07	0.04	0.03	0.04	0.07	0.04	0.03	0.01
16	0.01	0.04	0.06	0.08	0.08	0.05	0.04	0.04	0.09	0.06	0.03	0.02
18	0.02	0.05	0.07	0.09	0.10	0.06	0.05	0.05	0.10	0.07	0.04	0.02
20	0.02	0.06	0.08	0.11	0.12	0.06	0.06	0.06	0.11	0.09	0.05	0.03
22	0.02	0.07	0.09	0.12	0.14	0.07	0.07	0.07	0.13	0.10	0.06	0.03
24	0.03	0.08	0.10	0.14	0.15	0.08	0.08	0.08	0.14	0.11	0.07	0.04
26	0.03	0.09	0.11	0.15	0.16	0.08	0.09	0.08	0.15	0.12	0.07	0.04
28	0.04	0.11	0.12	0.17	0.18	0.09	0.10	0.09	0.17	0.14	0.08	0.05
30	0.05	0.12	0.13	0.19	0.20	0.10	0.11	0.10	0.19	0.15	0.09	0.06
32	0.06	0.13	0.14	0.20	0.21	0.11	0.12	0.11	0.20	0.16	0.10	0.07
36	0.07	0.15	0.16	0.22	0.23	0.12	0.13	0.12	0.22	0.19	0.11	0.08

## Data Availability

Data generated or analyzed during this study are available from the corresponding author upon reasonable request.

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
