# Peer review of "Comprehensive Monitoring Method for Diaphragm Wall Deformation Combining Distributed and Point Monitoring in Key Areas"

_sensors, 2025, doi:10.3390/s25072232_

Round 1
Reviewer 1 Report
Comments and Suggestions for Authors
This paper introduces a hybrid monitoring approach that combines Ultra-Weak Fiber Bragg Grating (UWFBG) technology with traditional FBG sensors to address the challenges of real-time, distributed deformation monitoring of diaphragm walls. Overall, the paper is well-structured and well-written. However, the following comments should be addressed before it can be considered for publication.
- What are the main challenges associated with traditional monitoring methods for diaphragm walls?
- How does the proposed hybrid monitoring method improve upon existing techniques?
- Can the authors explain the working principle of Fiber Bragg Grating (FBG) sensors and their role in this study?
- What advantages does Ultra-Weak Fiber Bragg Grating (UWFBG) technology provide in deformation monitoring?
- How did the strain-depth curve help in analyzing the structural behavior of the diaphragm wall?
- How did the monitoring results indicate the presence and progression of cracks in the concrete beams?
- What limitations or challenges were identified in the study?
Author Response
1.What are the main challenges associated with traditional monitoring methods for diaphragm walls?
Response: Traditional monitoring methods for diaphragm walls face challenges such as limited accuracy, low efficiency, susceptibility to environmental interference, and inability to provide real-time, distributed data coverage, often requiring costly equipment for advanced applications. For details, please refer to line 72 of the article.
2.How does the proposed hybrid monitoring method improve upon existing techniques?
Response: This method combines the UWFBG technology with traditional FBG technology, achieving a combination of distributed coverage and point-based precision. This method solved the shortcomings of traditional point-based monitoring, which has blind spots, and the limitation that distributed monitoring is difficult to economically monitor key areas in real time. For details, please refer to line 75 of the article.
3.Can the authors explain the working principle of Fiber Bragg Grating (FBG) sensors and their role in this study?
Response: Thank you for your valuable suggestions. In accordance with your request, we have provided a detailed explanation of the FBG working principle in section "2.1. FBG Sensing Principle" of the article.
4.What advantages does Ultra-Weak Fiber Bragg Grating (UWFBG) technology provide in deformation monitoring?
Response: UWFBG technology offers high sensitivity, distributed sensing, and immunity to elec-tromagnetic interference, along with long-term stability and low power consumption, making it ideal for large-scale, long-term monitoring. For details, please refer to line 136 of the article.
5.How did the strain-depth curve help in analyzing the structural behavior of the diaphragm wall?
Response: The spatial resolution of the optical cable actually used on site in this paper is 1 m, which can achieve the comparison of relative strain values within the 2 m length range of the structure. This can be unified with the strain distribution range of the beam in the indoor test. In addition, this paper uses optical cables sensitive to local cracks and high-strength optical cables (the same indoor and on-site) for the test results of the same quantity, and establishes the relationship between the strain level obtained by the high-strength optical cable and the crack appearance and development of the crack-sensitive optical cable. Finally, the strain data of the high-strength optical cable of the on-site ground-connected wall can be used to evaluate the local health status of the structure. In addition, the strain-depth curve revealed distinct deformation patterns by identifying strain concentration zones, enabling division of the wall into critical regions like the Upper Resistance Area (URA) and Main Load Area (MLA). This analysis clarified how excavation stages and strut placements influenced strain distribution, providing insights into crack risks and structural stability. For details, please refer to line 297 and line 325 of the article.
6.How did the monitoring results indicate the presence and progression of cracks in the concrete beams?
Response: The monitoring results detected strain peaks along the optical fiber, each corresponding to crack initiation. Crack progression was quantified by integrating strain peaks into widths, validated against crack width gauge measurements, with faster propagation rates observed in high-stress zones under distribution beams. In addition, by analyzing the relationship between crack development and pile full strain level, the possible development state of cracks in the on-site ground-connected wall was further evaluated. For details, please refer to line 190 of the article.
7.What limitations or challenges were identified in the study?
Response: The crack development index has not yet established a quantitative relationship with the strain data obtained by the optical cable, which will be further studied in subsequent studies. please refer to line 358.
Reviewer 2 Report
Comments and Suggestions for Authors
The manuscript presents a hybrid monitoring method for diaphragm wall deformation using a combination of ultra-weak Fiber Bragg Grating (UWFBG) technology and traditional FBG sensors. The study addresses an important topic in geotechnical engineering, and the approach proposed by the authors is innovative. However, the manuscript needs significant improvements before it can be considered for publication.
Some points of the study should be improved or better explained. A review of the manuscript is required.
- The introduction should provide a more extensive discussion of the state-of-the-art of fibre optic sensors in geotechnical applications.
- The references are geographically limited, with an overrepresentation of Chinese sources. It would be beneficial to cite research from other countries, such as the work of Prof. Lucio Olivares and Prof. Luigi Zeni (University of Campania "Vanvitelli"), Prof. Luca Schenato (University of Padova), Prof. Klar (Israel Institute of Technology ), who have conducted significant studies on fibre optic sensors for structural health monitoring.
- The description of the sensing principles in the manuscript is rather limited. The authors should provide a more detailed explanation of how Fiber Bragg Grating (FBG) and Ultra-Weak Fiber Bragg Grating (UWFBG) sensors operate. Specifically, it is important to clarify how the wavelength shifts in optical fibres are converted into strain measurements and how these measurements are used to infer structural deformations. Furthermore, a discussion on key aspects such as sensitivity, accuracy, potential sources of error, and external influencing factors (e.g., temperature, mechanical coupling) would enhance the understanding of the proposed monitoring approach.
- As for the beam experiment, the use of fibre optics appears appropriate for calibrating the system in relation to crack opening at a local scale. However, there is no global framework for the monitoring system, as there is a lack of data on the overall displacements obtained with traditional instrumentation (e.g., LVDTs). Including such data would provide additional insights into the global interpretation of the results.
- The comparison between fibre optic monitoring data and those obtained with crack width gauge should be presented to validate results obtained with the proposed system.
- The magnitude of strain in a structure is strictly related to the gradient of displacement, meaning that even small absolute displacements concentrated in a limited region can generate significant local strain values. Conversely, larger displacements that are distributed over a wider area may result in lower strain values. Since the manuscript lacks explicit data on the total displacements in both the beam experiment and the diaphragm wall case study, it remains unclear whether the observed strain values are directly comparable. The authors should explicitly discuss whether the observed strain values are directly comparable or justify why displacement measurements were not considered necessary.
- The discussion primarily provides a general overview of the proposed technology in terms of addressing the limitations of discrete monitoring systems, but a well-structured discussion of the obtained results should be formulated.
- The overcoming of traditional monitoring approaches should be discussed appropriately by comparing them with existing studies and highlighting the advantages of the proposed approach since distributed optic sensors have been extensively studied and widely implemented in the last decades.
- The figures should be improved for better readability. Some are difficult to interpret due to low resolution and unclear legends.
- Ensure proper numbering of tables and figures (e.g., "Table 5.4" appears without a preceding "Table 5.3").
- Figure 3 and Figure 16 need more evident labels and a more detailed description.
- Ensure that all references follow the same citation style.
Final Recommendation:
The manuscript presents an interesting approach but requires major revisions before it can be considered for publication. Addressing the issues outlined above will significantly enhance its scientific rigour and clarity.
Comments on the Quality of English LanguageThe manuscript contains grammatical errors and awkward phrasing. A comprehensive review of English is highly recommended.
Author Response
1.The introduction should provide a more extensive discussion of the state-of-the-art of fibre optic sensors in geotechnical applications.
The references are geographically limited, with an overrepresentation of Chinese sources. It would be beneficial to cite research from other countries, such as the work of Prof. Lucio Olivares and Prof. Luigi Zeni (University of Campania "Vanvitelli"), Prof. Luca Schenato (University of Padova), Prof. Klar (Israel Institute of Technology ), who have conducted significant studies on fibre optic sensors for structural health monitoring.
Response: Thank you for your suggestion. We have discussed the latest advancements in fiber optic sensors in the field of geotechnical engineering in the introduction section and have cited the works of the experts you mentioned. For details, please refer to lines 65-70 of the article.
2.The description of the sensing principles in the manuscript is rather limited. The authors should provide a more detailed explanation of how Fiber Bragg Grating (FBG) and Ultra-Weak Fiber Bragg Grating (UWFBG) sensors operate. Specifically, it is important to clarify how the wavelength shifts in optical fibres are converted into strain measurements and how these measurements are used to infer structural deformations. Furthermore, a discussion on key aspects such as sensitivity, accuracy, potential sources of error, and external influencing factors (e.g., temperature, mechanical coupling) would enhance the understanding of the proposed monitoring approach.
Response: Thank you for your valuable suggestion. We have provided a detailed description of the working principle of FBG in section 2.1 of the article.
3.As for the beam experiment, the use of fibre optics appears appropriate for calibrating the system in relation to crack opening at a local scale. However, there is no global framework for the monitoring system, as there is a lack of data on the overall displacements obtained with traditional instrumentation (e.g., LVDTs). Including such data would provide additional insights into the global interpretation of the results.
Response: This paper combines distributed strain measurement with local inclination measurement, uses UWFBG to obtain distributed strain data of underground continuous walls, and identifies local damage to the structure through strain singular points. The horizontal movement state of the wall is calculated by the data obtained from the FBG inclinometer array. This enables comprehensive monitoring of local damage and overall movement state of the underground continuous wall. The beam test is mainly to verify the feasibility of using strain data to identify local damage to the structure, so a comparative study of displacement data has not been carried out.
4.The comparison between fibre optic monitoring data and those obtained with crack width gauge should be presented to validate results obtained with the proposed system.
Response: Thank you for your valuable suggestion. We compare the crack meter data with the fiber optic data in Figure 9. We apologize for any misunderstanding caused by our description and have revised the annotations and description in Figure 9.
5.The magnitude of strain in a structure is strictly related to the gradient of displacement, meaning that even small absolute displacements concentrated in a limited region can generate significant local strain values. Conversely, larger displacements that are distributed over a wider area may result in lower strain values. Since the manuscript lacks explicit data on the total displacements in both the beam experiment and the diaphragm wall case study, it remains unclear whether the observed strain values are directly comparable. The authors should explicitly discuss whether the observed strain values are directly comparable or justify why displacement measurements were not considered necessary.
Response: Thank you for your valuable suggestion. The spatial resolution of the optical cable actually used on site in this paper is 1 m, which can achieve the comparison of relative strain values ​​within the 2 m length range of the structure. This can be unified with the strain distribution range of the beam in the indoor test. In addition, this paper uses optical cables sensitive to local cracks and high-strength optical cables (the same indoor and on-site) for the test results of the same quantity, and establishes the relationship between the strain level obtained by the high-strength optical cable and the crack appearance and development of the crack-sensitive optical cable. Finally, the strain data of the high-strength optical cable of the on-site ground-connected wall can be used to evaluate the local health status of the structure. In addition, the horizontal movement of the ground-connected wall causes possible accidents, and this paper uses FBG inclinometer data to achieve monitoring and evaluation.
6.The discussion primarily provides a general overview of the proposed technology in terms of addressing the limitations of discrete monitoring systems, but a well-structured discussion of the obtained results should be formulated.
Response: Thank you for your valuable suggestion. Related discussion has been added
7.The overcoming of traditional monitoring approaches should be discussed appropriately by comparing them with existing studies and highlighting the advantages of the proposed approach since distributed optic sensors have been extensively studied and widely implemented in the last decades.
Response: This paper combines distributed strain measurement with local inclination measurement, uses UWFBG to obtain distributed strain data of underground continuous walls, and identifies local damage to the structure through strain singular points. The horizontal motion state of the wall is calculated through the data obtained from the FBG inclinometer array, which can comprehensively monitor the local damage and overall motion state of the underground continuous wall. Compared with traditional point sensors, this paper can obtain the deformation state of the structure in a distributed manner and realize the identification of local damage. Related content is added in the text
8.The figures should be improved for better readability. Some are difficult to interpret due to low resolution and unclear legends.
Response: Thank you for your valuable suggestion. The relevant figures have been modified.
9.Ensure proper numbering of tables and figures (e.g., "Table 5.4" appears without a preceding "Table 5.3").
Response: Thank you for your valuable suggestion. In response to your concerns, we have thoroughly reviewed the figure and table titles and their numbering in the article.
10.Figure 3 and Figure 16 need more evident labels and a more detailed description.
Response: Thank you for your valuable suggestion. The relevant figures have been modified.
11.Ensure that all references follow the same citation style.
Response: Thank you for your valuable suggestions. We have conducted a thorough review of the reference formatting to ensure it aligns with the journal's requirements.
Reviewer 3 Report
Comments and Suggestions for Authors
Dear authors,
Your paper is of interest to geotechnical engineers but it actually lacks a lot of important informations. Your article should be self-consistent, i.e. it should contain all relevant information to understand your work.
First, the sensing principle of Ultra-Weak FBGs is not described. What is the purpose of using two Electro-Optic Modulators ? What is the spectral range covered by the tunable laser ? What are the repetition rate and acquisition rate ? Furthermore, what is the period between gratings ? Are the gratings photowritten at the same nominal wavelength ? According to Fig. 13, it seems to be one meter, please confirm. This is of fundamental importance for detecting strain peaks due to cracks.
Second, the basic principle of the FBG tiltmeter is not described either. Each tiltmeter includes two FBGs. It is a 1D tiltmeter (only one angle) ? Do they operate in push-pull configuration ? How many tiltmeters were used in the use-case ?
Tiltmeters provide angular data from which the strain may be inferred by a first-order integration. FBG sensors provide direct strain measurement, delivering an estimation of the curvature radius. It is not clear in the paper how those data are merged to improve the decision-making. Please elaborate a bit more about this.
The whole paper is rather well-written and the English is satisfactory. Please correct for the error in many figures (Fig. 2, 8 and 10) : one should read 'tilt sensors' instead of 'title sensors'.
Page 2, line 55 : 'anti-interference' is an improper term. Is is electromagnetic immunity ?
Page 11, line 264 : 800 microstrain.
Author Response
Your paper is of interest to geotechnical engineers but it actually lacks a lot of important informations. Your article should be self-consistent, i.e. it should contain all relevant information to understand your work.
1.First, the sensing principle of Ultra-Weak FBGs is not described. What is the purpose of using two Electro-Optic Modulators ? What is the spectral range covered by the tunable laser ? What are the repetition rate and acquisition rate ? Furthermore, what is the period between gratings ? Are the gratings photowritten at the same nominal wavelength ? According to Fig. 13, it seems to be one meter, please confirm. This is of fundamental importance for detecting strain peaks due to cracks.
Response: In response to your concerns, we have described the sensing principle of Ultra-Weak FBG in section "2.2. Ultra-Weak FBG Technology " of the article, as detailed on line 122.
2.Second, the basic principle of the FBG tiltmeter is not described either. Each tiltmeter includes two FBGs. It is a 1D tiltmeter (only one angle) ? Do they operate in push-pull configuration ? How many tiltmeters were used in the use-case ?
Response: The FBG tiltmeter operates as a 1D sensor measuring inclination in a single plane using two FBGs arranged in a push-pull configuration. The dual-FBG design compensates for temperature effects and enhances sensitivity by correlating wavelength shifts from opposing strain states (one FBG under tension, the other under compression). In the case study, two tiltmeters were deployed at depths of 7 m and 25 m (Fig. 8), enabling real-time angular displacement monitoring at critical locations of the diaphragm wall.
3.Tiltmeters provide angular data from which the strain may be inferred by a first-order integration. FBG sensors provide direct strain measurement, delivering an estimation of the curvature radius. It is not clear in the paper how those data are merged to improve the decision-making. Please elaborate a bit more about this.
Response: This paper combines distributed strain measurement with local inclination measurement, uses UWFBG to obtain distributed strain data of underground continuous walls, and identifies local damage to the structure through strain singular points. The horizontal motion state of the wall is calculated through the data obtained from the FBG inclinometer array, which can comprehensively monitor the local damage and overall motion state of the underground continuous wall. Related content is added in the text.
4.The whole paper is rather well-written and the English is satisfactory. Please correct for the error in many figures (Fig. 2, 8 and 10) : one should read 'tilt sensors' instead of 'title sensors'.
Response: Thank you for your careful checking, we will correct the errors in the figures.
5.Page 2, line 55 : 'anti-interference' is an improper term. Is is electromagnetic immunity ?
Response: Thank you for your careful checking, we will correct the word.
6.Page 11, line 264 : 800 microstrain.
Response: Thank you for pointing out the issue. We have made the changes as per your request.
Round 2
Reviewer 1 Report
Comments and Suggestions for Authors
The paper is accepted in present form.
Reviewer 2 Report
Comments and Suggestions for Authors
The authors have incorporated the reviewer’s suggestions. The manuscript may be accepted in its current form.